# Recombinant *Lactobacillus casei* Expressing Capsid Protein VP60 can Serve as Vaccine Against Rabbit Hemorrhagic Disease Virus in Rabbits

**DOI:** 10.3390/vaccines7040172

**Published:** 2019-11-02

**Authors:** Li Wang, Tian Xia, Tiantian Guo, Yi Ru, Yanping Jiang, Wen Cui, Han Zhou, Xinyuan Qiao, Lijie Tang, Yigang Xu, Yijing Li

**Affiliations:** 1College of Veterinary Medicine, Northeast Agricultural University, Harbin 150030, China; wanglicau@163.com (L.W.); 13206521680@163.com (T.X.); 15645171752@163.com (T.G.); jiangyanping2017@126.com (Y.J.); cuiwen_200@163.com (W.C.); zhouhan9659@163.com (H.Z.); qiaoxinyuan@126.com (X.Q.); tanglijie@neau.edu.cn (L.T.); 2State Key Laboratory of Veterinary Etiological Biology, Lanzhou Veterinary Research Institute, Chinese Academy of Agricultural Sciences, Lanzhou 730000, China; ruyi@caas.cn; 3China Ministry of Agriculture Key Laboratory of Animal Pathogen Biology, Northeastern Science Inspection Station, Harbin 150030, China

**Keywords:** *Lactobacillus casei* (*L. casei*), rabbit hemorrhagic disease virus (RHDV), vaccine, VP60(VP1)

## Abstract

Rabbit hemorrhagic disease virus (RHDV) is the causative agent of rabbit hemorrhagic disease (RHD). RHD, characterized by hemorrhaging, liver necrosis, and high morbidity and mortality in rabbits and hares, causes severe economic losses in the rabbit industry worldwide. Due to the lack of an efficient in-vitro propagation system for RHDV, the current vaccine is produced via chemical inactivation of crude RHDV preparation derived from the livers of infected rabbits. Inactivated vaccines are effective for controlling RHD, but the potential problems of biosafety and animal welfare have negative effects on the application of inactivated vaccines. In this study, an oral Lactobacillus casei (*L. casei*) vaccine was used as an antigen delivery system to express RHDV capsid protein VP60(VP1)-eGFP fusion protein. The expression of the recombinant protein was confirmed via western blotting and immunofluorescence (IFA). Our results indicate that oral administration of this probiotic vaccine can stimulate secretory immunoglobulin A (SIgA)-based mucosal and IgG-based humoral immune responses in rabbits. The immunized rabbits were completely protected against challenge with RHDV. Our findings indicate that the *L. casei* expression system is a new strategy for the development of a safe and efficient vaccine against RHDV.

## 1. Introduction

Rabbit hemorrhagic disease (RHD), characterized by severe necrotizing hepatitis and disseminated intravascular coagulation in the liver, spleen, kidney and other solid organs, is a highly contagious and lethal infection in rabbits [1,2]. RHD is peracute and often lethal hepatitis caused by the rabbit caliciviruses *Lagovirus europaeus* GI.1 (previously called rabbit hemorrhagic disease virus—RHDV) and *Lagovirus europaeus* GI.2 (previously called RHDV2 or RHDVb) [1,3,4]. This disease was first reported in China in 1984, and has, since, spread rapidly around the world in less than 10 years, causing considerable economic losses in the rabbit industry and impacting the ecology of wild rabbit populations [1,2,5,6]. In 1989, OIE designated this illness as a viral hemorrhagic disease and added it to List B of the International Animal Health Code [7]. Subsequently, a novel lagovirus, GI.2, emerged in France in 2010 [4]. GI.2 is now endemic in Europe and Australia, and appears to be replacing GI.1 strains in these regions [8,9,10].

RHDV (rabbit hemorrhagic disease virus) is the etiologic factor of RHD. RHDV is a calicivirus in the genus Lagovirus, family Caliciviridae [1]. RHDV is a non-enveloped icosahedral virus possessing a single-stranded positive sense RNA genome that is approximately 7.4 kb in length [2,11]. The genome comprises a 5′ untranslated region (UTR), a 3′ UTR, and two overlapping open reading frames (ORFs): ORF1 and ORF2. ORF1 encodes a polyprotein that is cleaved by the viral protease into seven nonstructural proteins (NSP1-7) and a major structural capsid protein VP60 (presently VP1) at its C-terminus. VP10 (presently VP2) is a minor structural protein that is encoded by ORF2 [11,12,13].

Vaccination is the main approach for controlling RHDV because no effective treatment is available for this disease. Inactivated vaccines against RHDV were introduced in the early 1990s, improving the survival of rabbits on rabbit farms [7,14]. However, RHDV inactivated vaccines are manufactured using the livers of rabbits infected with RHDV. This is because RHDV cannot grow in any continuous cell lines. Therefore, biological risks, animal-welfare concerns, and high costs are the major bottleneck problems in the production and usage of tissue-inactivated vaccines [2].

RHDV spreads mainly through the upper respiratory and digestive tracts. The initial steps leading to RHDV infection take place on mucosal surfaces [15,16]. It is generally believed that mucosal immunization is an effective approach for preventing systemic infection by pathogens present on mucosal surfaces [17]. The gastrointestinal (GI) tract is the largest mucosal surface accessible via oral administration [18]. Oral vaccination can trigger a response involving neutralizing mucosal antibodies (IgA) and cell-mediated immunity, and does not interfere with IgG-based responses [19,20,21,22]. Additionally, oral vaccines show better safety and compliance profiles, and are simpler to manufacture and administer, than traditional injectable preparations [23,24]. However, the delivery of antigens for oral vaccination of the GI tract is hindered by multiple physicochemical and biological barriers; antigens can be subjected to early disintegration and advanced degradation by low pH and proteases present in the GI tract [25,26]. *L. casei* is a probiotic that is well known for its health-promoting properties, such as maintaining homeostasis and suppressing pathogens in humans and animals [27]. *L. casei* has shown a good safety profile, can colonize the intestine, and exerts a nonspecific immunoadjuvant effect. For those reasons, oral vaccines using *L. casei* as a delivery system for pathogenic antigens have garnered much interest in vaccine development [27,28,29]. Currently, there is increasing interest in the development of *L. casei* oral vaccines, and this approach is significant for the effective induction of a mucosal immune response [30]. The results to date have been confirmed that the safety and the effectiveness of *L. casei* were used as the oral vaccine vehicle, which were extensively used in protecting individuals against a variety of pathogens [31,32,33,34,35,36,37].

Developing an efficient and safe oral vaccine that can induce strong mucosal and systemic immune responses is desirable for effective prevention of RHDV. Therefore, in our current study, we developed a recombinant *L. casei* expressing the major structural capsid protein VP60(VP1)-eGFP fusion protein of RHDV. Then we evaluated the humoral and mucosal immune responses to this recombinant *L. casei*, and assessed its immunogenic properties upon administration as an oral vaccine.

## 2. Materials and Methods

### 2.1. Bacterial Strain, Virus, Primers, and Plasmid

Lactobacillus casei ATCC 393 (LC393), kindly provided by Jos Seegers (NIZO, Ede, The Netherlands), the vector pPG-T7g10-PPT transformed into competent LC393 cells by electroporation to generate L.casei pPG-T7g10 (pPG/LC393). Lactobacillus casei was cultured in de Man, Rogosa and Sharp (MRS) broth at 37 °C without shaking under anaerobic conditions. RHDV-1 was extracted from homogenates of liver samples obtained from RHDV-infected rabbits in our laboratory. The sequences of the primers used for PCR amplification of VP60 were as follows: 5′-ATGGAGGGCAAAACCCGCACAGCGC-3′; 5′-TCAGACATAAGAAAAGCCATTGGCT-3′. pPG-T7g10-PPT, constructed in our previous study [33], was used as a constitutive expression plasmid in this study; it encoded resistance to chloramphenicol (Cm) and contained an HCE promoter, T7g10 enhancer, PgsA anchor, and the rrnBT1T2 terminator. Plasmids pMD18-T simple-eGFP and pMD18-T simple-VP60 were constructed in our laboratory previously.

### 2.2. Construction of Recombinant L. casei (pPG-T7g10-eGFP-VP60/LC393)

A schematic diagram of the recombinant DNA plasmid is shown in Figure 1. In brief, recombinant plasmid pMD18-T simple-eGFP was cleaved with SacI and KpnI restriction endonuclease and inserted into the corresponding sites of a pMD18-T simple-VP60 expression vector digested using SacI and KpnI. This generated recombinant plasmid pMD18-T simple-eGFP-VP60. The eGFP-VP60 fragment, cleaved with SacI and ApaI, was inserted into a constitutive expression vector pPG-T7g10-PPT, resulting in the recombinant plasmid pPG-T7g10-eGFP-VP60 (Cm^+^). Afterwards, the Cm resistance gene in pPG-T7g10-eGFP-VP60 (Cm^+^) was disrupted using MunI and NcoI. Finally, the recombinant plasmid pPG-T7g10-eGFP-VP60 was obtained via blunt-end ligation and transformed into competent LC393 cells by electroporation to generate recombinant *L. casei* pPG-eGFP-VP60/LC393.

### 2.3. Analysis of Protein Expression by Western Blotting

The recombinant L. casei pPG-eGFP-VP60/LC393, pPG/LC393, and LC393 were cultured overnight in MRS broth and harvested by centrifugation at 9000 × *g* for 10 min at 4 °C After cell lysis and centrifugation, the supernatant was subjected to 12% sodium dodecyl sulfate-polyacrylamide gel electrophoresis (SDS-PAGE) and western blot assay. The proteins in the supernatant were separated by SDS-PAGE and electrotransferred onto PVDF membranes (Millipore, Milford, MA, USA). The immunoblot was then incubated with the following primary antibodies: Rabbit polyclonal anti-VP60(VP1) prepared in our laboratory (at the dilution of 1:200), and mouse monoclonal anti-eGFP (at the dilution of 1:5000) (Abcam, Cambridge, MA, USA) eat 4 °C overnight. Then, the immunoblot was incubated with horseradish peroxidase (HRP)-conjugated goat anti-rabbit or goat anti-mouse IgG secondary antibodies (at the dilution of 1:5000) (Sigma, Ronkonkoma, NY, USA) at 4 °C for 30 min. Protein expression was visualized using a chemiluminescent substrate reagent (Thermo Scientific, Durham, NC, USA) according to the manufacturer’s instructions.

### 2.4. Analysis of Protein Expression by Indirect Immunofluorescence

We used fluorescence microscopy and indirect immunofluorescence assay (IFA) to detect the expression of VP60 (VP1) and eGFP. In brief, recombinant *L. casei* was cultured overnight in MRS at 37 °C. Cell pellets were washed thrice with PBS, incubated with rabbit anti-VP60(VP1) hyper-immune serum(prepared in our laboratory), and then with TRITC red fluorescently labeled anti-rabbit IgG secondary antibody (diluted at 1:200) at 37 °C for 30 min. Cell pellets were then washed thrice, resuspended in 200 μL PBS, and smeared onto microscope slides. Images were obtained via laser confocal microscopy (model LSM510 META; Zeiss, Oberkochen, Germany).

### 2.5. Immunization, Specimen Collection, and RHDV Challenge

Two-month-old specific pathogen-free (SPF) rabbits, obtained from Liaoning Changsheng Biotechnology Co., Ltd. (Liaoning, China) were housed under SPF conditions with free access to standard chow diet and water. Prior to oral administration, recombinant L. casei was cultured overnight in MRS medium, washed with PBS, and then resuspended in PBS at a final concentration of 1 × 10^10^ CFU/mL. The rabbits were randomly separated into three groups, with six rabbits per group.

The first group was injected with a single does of a commercial inactivated RHDV vaccine. The second group was orally administered 5 mL *L. casei* pPG-eGFP-VP60/LC393 (1 × 10^10^ CFU/mL). The third group, serving as unimmunized control, was inoculated with 5 mL of orally administered PBS. The rabbits were immunized orally once a day for three consecutive days and boosted twice at 2-week intervals (Figure 2).

Serum, nasal washes, vaginal lotions, tears, and fecal samples were obtained on days 0, 7, 14, 21, 28, and 35 after immunization, and were stored at −20 °C until use. Fecal samples were pretreated according to a previously published method [35].

Three groups of rabbits (*n* = 6), immunized with pPG-eGFP-VP60/LC393, inactivated vaccine, or PBS, were challenged with 1 mL RHDV crude liver extract (at a viral RNA copy number of 1.0 × 10^5.79^) administered via intramuscular injection into the leg at 37 d after the first vaccination (Figure 2). The viral RNA copy number was determined by Real-time PCR. All rabbits were humanely euthanized and dissected at 10 days post-infection (DPI).

### 2.6. ELISA Immunoassay

The levels of anti-VP60(VP1)-specific IgG in the sera, and those of SigA in the feces, genital tract secretions, and intestinal mucusa, of the rabbits used in this study, were determined via ELISA. Briefly, polystyrene microtiter plates were coated with RHDV and incubated overnight at 4 °C. After washing thrice with PBST (PBS containing 1% Tween-20), the plates were blocked with 5% skimmed milk at 37 °C for 2 h. After washing thrice with PBST, the collected samples, prepared in triplicate, were diluted with PBS (sera diluted at 1:50; supernatants of feces, tears, vagina, and nasal cavity lavage diluted at 1:10), incubated at 37 °C for 1 h. Then, an HRP-conjugated goat anti-rabbit IgG or IgA antibody (Sigma, Ronkonkoma, NY, USA) was added to each well (at 1:2000), and incubated for 1 h at 37 °C. Color was developed using tetramethylbenzidine (TMB) (Sigma, Ronkonkoma, NY, USA) as a substrate, and absorbance was measured at OD 490.

### 2.7. Histopathological Analysis

The liver, spleen, and small intestine obtained from the rabbits were subjected to histopathological examination. Tissue samples were fixed in 10% neutral-buffered formalin, processed routinely, embedded into paraffin blocks, and sectioned at 4 μm. The sections were stained with hematoxylin and eosin (H&E), and morphology was evaluated using light microscopy.

### 2.8. Statistical Analysis

All data are presented as mean ± standard deviation (SD). The samples from each group (6 rabbits) were pooled and then analyzed as one sample in triplicate. Data were analyzed using two-way ANOVA with multiple comparison (LSD) test in SPSS. Different letters (a vs. b, a vs. c, b vs. c) indicate significant differences (*p* < 0.01) at the same time point.

## 3. Results

### 3.1. Determination of Chloramphenicol Resistance and Inherited Stability of Recombinant L. casei

The recombinant plasmid pPG-T7g10-eGFP-VP60 was generated (Figure 1) and transformed by electroporation into competent LC393 cells. eGFP-positive recombinant *L. casei* cells were collected by flow cytometry and designated as pPG-eGFP-VP60/LC393. To assess whether recombinant *L. casei* retained chloramphenicol resistance, recombinant *L. casei* were cultivated on MRS solid plates containing and not containing chloramphenicol. Our results indicate that recombinant *L. casei* pPG-eGFP-VP60/LC393 grew normally on MRS medium without chloramphenicol, but did not grow on MRS medium containing chloramphenicol. This indicates that the chloramphenicol gene had been removed successfully (Figure 3a).

We then examined the inherited stability of the recombinant *L. casei*. Recombinant *L. casei* pPG-eGFP-VP60/LC393 was serially inoculated 40 times. The 5th, 10th, 15th, 20th, 25th, 30th, 35th, and 40th generations of recombinant *L. casei* were cultured for 16 h, and the plasmid was extracted. Inherited stability was then assessed using PCR and sequencing. Our results indicate that recombinant *L. casei* showed good inherited stability and no gene deletion (Figure 3b).

### 3.2. Expression of the RHDV VP60(VP1) by Recombinant L. casei

Recombinant *L. casei* pPG-eGFP-VP60/LC393, pPG/LC393, and LC393 cells were cultured in MRS broth without chloramphenicol antibiotic. Cell lysates were analyzed by western blotting using mouse anti-eGFP monoclonal antibody and rabbit anti-VP60(VP1) polyclonal antibody. For both anti-eGFP and anti-VP60(VP1) antibodies, specific immunoreactive bands of the expected size were observed in recombinant *L. casei* pPG-eGFP-VP60/LC393, and there was no band at the corresponding position in pPG/LC393 or LC393 (Figure 4a). Confocal microscopy and IFA were used to determine the expression of the eGFP-VP60(VP1) fusion protein in recombinant *L. casei*. Confocal microscopy was also used to observe the expression of eGFP in *L. casei*. Our results show the emission of green fluorescence in *L. casei* pPG-eGFP-VP60/LC393, but not in pPG/LC393 (Figure 4b, top). For IFA, pPG-eGFP-VP60/LC393 and pPG/LC393 were incubated with rabbit anti-VP60(VP1) polyclonal antibody and TRITC-conjugated goat anti-rabbit IgG secondary antibody, and then observed using laser confocal microscopy. Emission of red fluorescence was observed in pPG-eGFP-VP60/LC393, but not in pPG/LC393 (Figure 4b, bottom). These results indicate that the protein of interest was successfully expressed in the recombinant *L. casei*.

### 3.3. Induction of Systemic and Mucosal Antibody Responses

Next, we analyzed the immunogenicity of recombinant *L. casei* pPG-eGFP-VP60/LC393 and effectiveness of the inactivated virus vaccine in rabbits after oral and intramuscular immunization. Systemic and mucosal immune responses were assessed by detecting the presence of anti-RHDV IgG and IgA antibodies; the levels of these antibodies were quantified using ELISA. Starting at day 7 after primary vaccination, a significantly increased level of anti-VP60(VP1)-specific IgG was detected in rabbits orally administered with pPG-eGFP-VP60/LC393, and in those injected with the inactivated vaccine, compared with the responses of the PBS-treated control group (*p* < 0.01). Additionally, there was less IgG present for the L. casei vaccine in comparison to the inactivated vaccine (Figure 5). Similarly, localized levels of anti-VP60(VP1)-specific SigA in the feces, tears, vagina, and nasal cavity lavage of the immunized rabbits were measured via ELISA. In brief, mucosal SigA levels increased significantly in the feces (Figure 6a), vagina (Figure 6b), nasal cavity (Figure 6c), and tears (Figure 6d) of the group orally immunized with pPG-eGFP-VP60/LC393 compared with those of the PBS-treated control group (*p* < 0.01). The level of SigA induced by oral immunization with pPG-eGFP-VP60/LC393 was higher than that induced by the inactivated vaccine.

### 3.4. Immune Protection Against RHDV Challenge

To assess immunoprotective efficacy, the three groups of rabbits administered pPG-eGFP-VP60/LC393, inactivated vaccine, or PBS were infected with 1 mL RHDV (RNA copy number of 1.0 × 10^5.97^) at 7 days after the last booster. Our results show that all rabbits in the PBS group died within three days post-challenge, while the rabbits immunized with recombinant *L. casei* or inactivated vaccine were completely protected against challenge with RHDV (Figure 7).

### 3.5. Histopathological Analysis on Different Tissues of the Infected Rabbits

The liver, spleen, and small intestine, collected from the rabbits in different treatment groups, were subjected to histopathological evaluation. Lesions on liver, spleen, and small intestine were more severe in the PBS-treated group than in the vaccinated groups. In the PBS-treated group, the hepatocytes in the liver presented a disordered appearance and clearly manifested degeneration and necrosis. Splenic corpuscles appeared destroyed, and some had disappeared entirely, splenic sinus was congested, and intestine villi were necrotic and severely desquamated. There are almost no differences between the pPG-eGFP-VP60/LC393-immunized group and inactivated vaccine-immunized group (Figure 8).

## 4. Discussion

Rabbit hemorrhagic disease (RHD) is a comparatively new and economically important viral disease that affects adult rabbits [38]. It is important to develop effective strategies for preventing and controlling the spread of RHDV. However, because there is no efficient in-vitro system for propagating RHDV, it is difficult to produce this virus as a source of vaccine antigens [39]. At present, most commercial vaccines are produced using viral preparations obtained from the livers of experimentally infected rabbits [40]. This strategy raises numerous concerns with respect to animal welfare and biological safety [2]. To address these issues, and to develop an effective, safe, and convenient oral vaccine against RHDV, we constructed a recombinant *L. casei* expressing the RHDV VP60(VP1) protein.

Antigens are key factors in vaccine development. Previous studies have shown that capsid protein VP60(VP1), a major immunogenic protein of RHDV, induces expression of a neutralizing antibody [41]. Rabbits inoculated with a sufficient amount of recombinant VP60(VP1) are fully protected from wild-type RHDV, and VP60 is commonly used to develop recombinant engineered vaccines against RHDV [41,42]. Therefore, VP60(VP1) was selected as an immunogen for the development of a recombinant engineered vaccine in this study. Recombinant engineered vaccines containing VP60(VP1) have been developed using bacteria, yeasts, plants, poxvirus-based vectors, and insect cells as delivery vectors [41,43,44,45]. However, intramuscular injection, which is the main route of administering these recombinant engineered vaccines, requires trained personnel, generates a vast amount of biohazardous waste, and can elicit a stress response in the animals [18,46]. These issues can be readily solved by live vector-based oral vaccines [18].

As an oral vaccine vector, *L. casei* offers many advantages, such as increased safety, resistance to the harsh GI environment, robust capacity for intestinal colonization, and nonspecific immunoadjuvant effect [47,48,49]. Previously, live vector-based oral vaccines were constructed using LC393, a promising strain of *L. casei* that can deliver antigenic material to the mucosa and express numerous targeted proteins, such as VP2 protein of porcine parvovirus (PPV), glycoprotein E2 of bovine viral diarrhea virus (BVDV), spike protein (S) of transmissible gastroenteritis virus (TGEV), and core neutralizing epitope (COE) antigen of porcine epidemic diarrhea virus (PEDV) [35,50,51,52]. Oral vaccination with these vaccines can efficiently induce mucosal, humoral, and cellular immune responses, indicating that LC393 is an excellent vector for use in oral vaccines. Therefore, LC393 was selected as a vector in the genetically engineered *L. casei* oral vaccine developed in our present study.

The main disadvantage of traditional plasmid expression systems is using antibiotic resistance genes as selective markers for genetically engineered bacteria [53,54]. Although antibiotics are helpful for vector screening, using antibiotic resistance genes in practice can impact biosafety, contribute to antibiotic resistance, and present potential hazards to health and the environment [55]. To avoid potential issues caused by antibiotic resistance markers, it is important to establish expression vectors without the use of antibiotic resistance genes. eGFP has many advantages as a genetic marker and reporter gene. eGFP expression is mostly non-toxic to cells, and eGFP fluorescence can last for an extended time [55]. In the present study, we constructed a recombinant *L. casei* pPG-T7g10-eGFP-VP60/LC393 using eGFP as a selection marker and removed chloramphenicol resistance from the vector. Analysis of inherited stability of pPG-T7g10-eGFP-VP60/LC393 demonstrated that recombinant *L. casei* was stably inherited for more than 40 generations with no gene deletion. We designed the pPG-T7g10-PPT expression vector, a constitutive cell-surface expressed plasmid, to construct recombinant *L. casei* for the delivery of vaccine antigen. The pPG-T7g10-PPT vector contained a T7g10 enhancer, derived from gene 10 of bacteriophage T7; the T7g10 enhancer can augment protein translation and expression [56]. Expression of the eGFP-VP60(VP1) fusion protein was confirmed via western blotting and IFA. Our results indicate that a genetically engineered oral vaccine against RDHV, using *L. casei* without antibiotic resistance gene, was developed successfully.

An effective oral vaccine should induce both systemic and mucosal immune responses. Thus, we evaluated the immunogenicity of orally administered pPG-eGFP-VP60/LC393 in rabbits. The multiple immunization procedure of recombinant Lactobacillus was employed in this study was due to the amounts of recombinant Lactobacillus adhered to the intestinal mucosa were decreased with the time [57]. The immune effect of the multiple immunization procedure of recombinant Lactobacillus has been confirmed in the previous studies [31,35]. The rabbits were immunized with inactivated vaccine according to the instruction. Serum-derived IgG can appreciably contribute to immune defense by reducing the ability of pathogens to cross intestinal mucosa, thereby limiting the systemic spread of invasive organisms [58]. We detected substantially increased levels of anti-RHDV IgG in rabbits orally immunized with pPG-eGFP-VP60/LC393; these levels increased rapidly following booster immunization. sIgA, a major immunoglobulin in mucosal immune responses, protects against pathogen invasion at mucosal sites [59]. Therefore, sIgA levels can be used to evaluate the protective efficacy of vaccines [60]. In the present study, we detected high levels of antigen-specific mucosal sIgA in the nasal washes, vaginal lotions, tears, and feces of rabbits orally immunized with pPG-eGFP-VP60/LC393. Recombinant *L. casei* pPG-eGFP-VP60/LC393 induced a higher level of sIgA compared with that induced by the inactivated virus vaccine, indicating that *L. casei* pPG-eGFP-VP60/LC393 was more efficient at eliciting mucosal immune responses, and there were almost no significant differences of sIgA and IgG antibody levels in the experimental animals administered with pPG/L. casei 393 and PBS, which have been confirmed in our previous studies [31,35,51]. Previous studies also have shown that oral vaccines elicit stronger mucosal immune responses than do injected vaccines [46]. In our present study, we have shown that oral vaccination with recombinant *L. casei* pPG-eGFP-VP60/LC393 can strongly induce anti-RHDV systemic and mucosal immune responses.

In this study, the protective efficacy of *L. casei* was analyzed using rabbits. Our results show that survival rates of rabbits vaccinated with recombinant *L. casei* and inactivated virus vaccine were at 100% after the viral challenge; however, none of the rabbits in the PBS-treated group survived challenge with RHDV. This finding indicates that pPG-eGFP-VP60/LC393, administered via oral immunization, effectively induced a sufficient immune response to protect rabbits against infection with RHDV. Rabbits infected with RDHV often develop acute necrotizing hepatitis and disseminated intravascular coagulation in the liver, spleen, kidney, and other solid organs [1]. Histopathological analysis showed topical histopathological changes in RDH of the liver and spleen of the PBS-treated group after challenge with RHDV. However, these changes were not observed in the organs of rabbits immunized with recombinant *L. casei* pPG-eGFP-VP60/LC393 or inactivated virus vaccine. The villi of the small intestine were necrotic and severely desquamated in the PBS-treated group and the group treated with inactivated virus vaccine. The group administered *L. casei* pPG-eGFP-VP60/LC393 showed the intact structure of the villi in the small intestine. This finding indicates that *L. casei* protected the intestinal environment against the effects of RHDV.

## 5. Conclusions

In conclusion, we developed an anti-RHDV vaccine for oral administration using *L. casei* to deliver the VP60(VP1) antigen of RHDV. Our findings show that genetically engineered recombinant *L. casei* pPG-eGFP-VP60/LC393 without antibiotic resistance could efficiently induce humoral and mucosal immune responses, thereby protecting rabbits from infection with RHDV.

## Figures and Tables

**Figure 1 vaccines-07-00172-f001:**
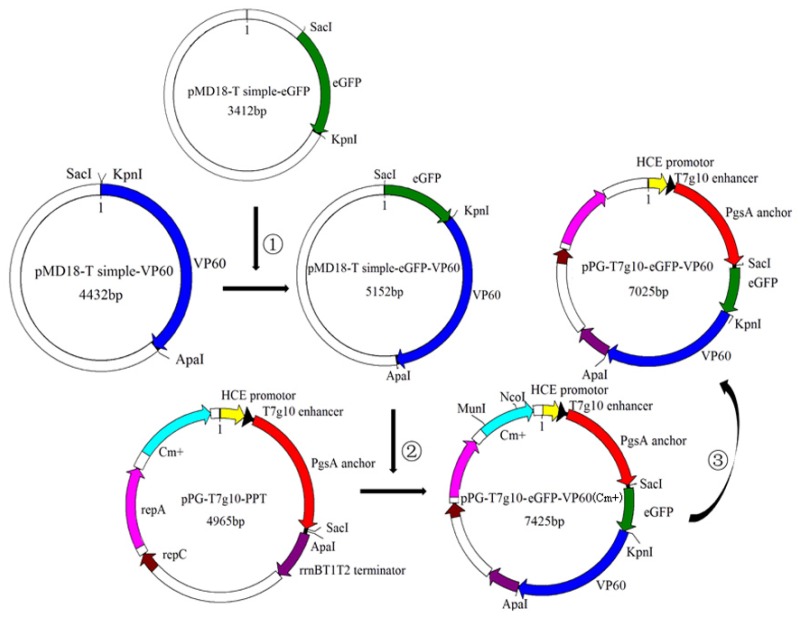
Schematic diagram showing the construction of recombinant plasmids. ①: eGFP, cleaved with *Sac*I and *Kpn*I, was inserted into the corresponding sites of pMD18-T simple-VP60 digested by *Sac*I and *Kpn*I; ②: eGFP-VP60 fragment, cleaved with *Sac*I and *Apa*I, was inserted into constitutive expression vector pPG-T7g10-PPT; ③: *Cm* resistance gene in pPG-T7g10-eGFP-VP60 (Cm^+^) was disrupted by *Mun*I and *Nco*I, and recombinant plasmid pPG-T7g10-eGFP-VP60 was obtained by blunt end ligation.

**Figure 2 vaccines-07-00172-f002:**
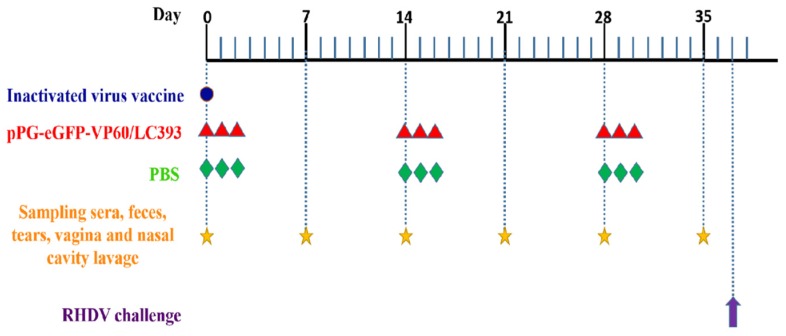
Immunization, viral challenge, and sampling. Blue dot indicates the time points used for administration of inactivated-virus vaccine; red triangles indicate the time points used for administration of recombinant *L. casei*; green diamonds indicate the time points used for administration of PBS; yellow stars indicate the time points used to collect samples, and purple arrow indicates the time points used for viral infections.

**Figure 3 vaccines-07-00172-f003:**
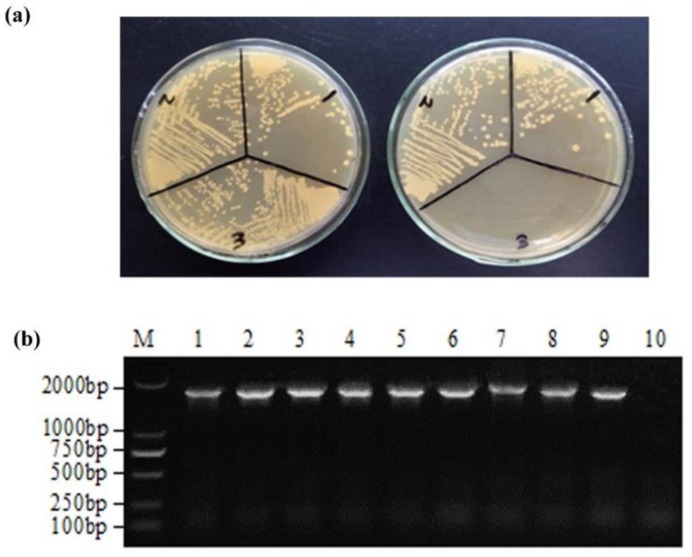
Identification of recombinant *L. casei* pPG-eGFP-VP60/LC393 (**a**) Recombinant *L. casei* cells were cultivated on MRS solid plates with (right) and without chloramphenicol (left). 1: pPG/LC393, 2: pPG-eGFP-VP60/LC393 (Cm^+^), 3: pPG-eGFP-VP60/LC393; (**b**) Inherited stability of recombinant *L. casei* pPG-eGFP-VP60/LC393. Recombinant *L. casei* pPG-eGFP-VP60/LC393 was serially inoculated 40 times. The plasmid was then extracted, and inherited stability was evaluated by PCR amplification of VP60. Lanes 1-8 indicate the 5th, 10th, 15th, 20th, 25th, 30th, 35th and 40th generations of recombinant *L. casei*; Lane 9 indicates positive control; Lane 10 indicates negative control; Lane M indicates DNA molecular weight markers.

**Figure 4 vaccines-07-00172-f004:**
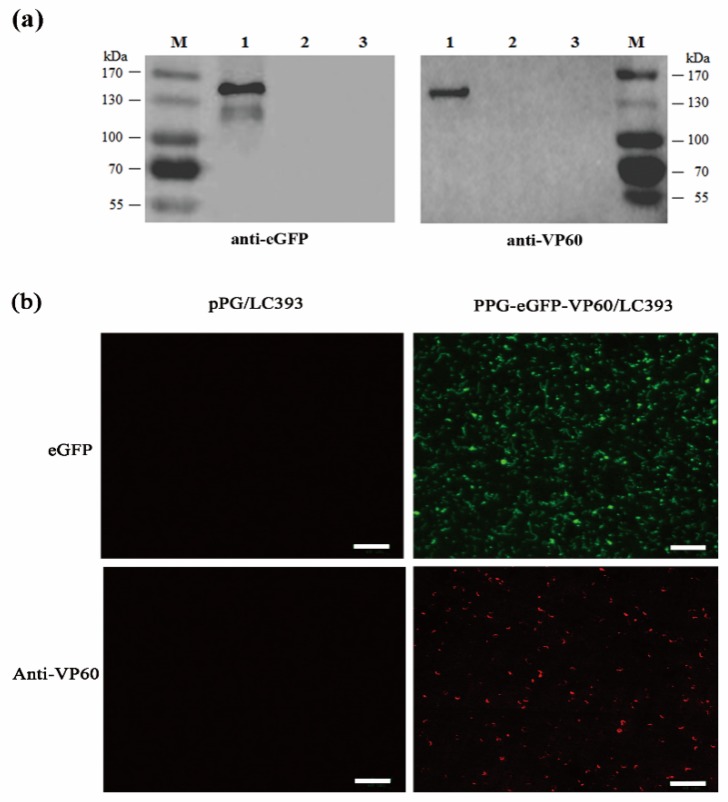
Identification of proteins of interest expressed in recombinant *L. casei* pPG-eGFP-VP60/LC393. (**a**) Identification of the eGFP-VP60(VP1) fusion protein was conducted by western blotting using anti-eGFP antibody (left) and anti-VP60(VP1) antibody (right). The relevant immunoreactive bands were detected in cell pellets of strain pPG-eGFP-VP60/LC393 (lane 1), but not those of strains pPG/LC393 (lane 2) or LC393 (lane 3). M: Protein molecular weight markers. (**b**) The fluorescence of eGFP was observed by confocal microscopy directly (top), identification of VP60 by IFA assay using an anti-VP60(VP1) antibody (bottom), respectively. Red and green fluorescence were observed in strain pPG-eGFP-VP60/LC393 (right), but not in strain pPG/LC393 (left). Scale bar represents 30 µm.

**Figure 5 vaccines-07-00172-f005:**
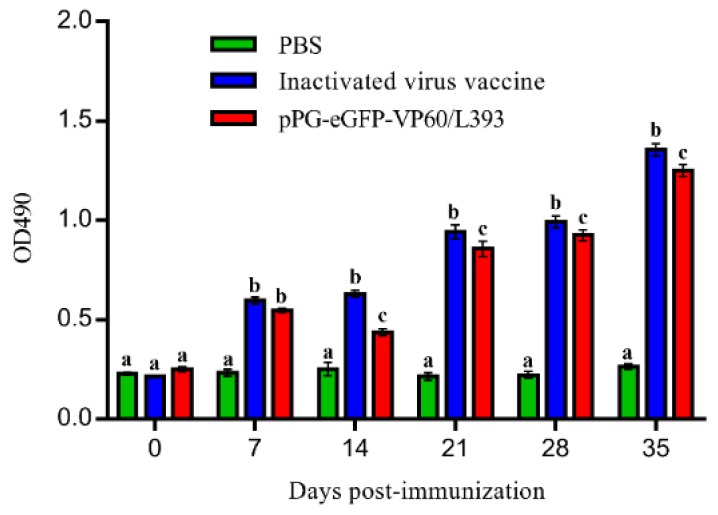
Levels anti-RHDV (rabbit hemorrhagic disease virus)-specific IgG in the sera of immunized rabbits. Levels of anti-RHDV-specific IgG in rabbit sera were quantified by ELISA using RHDV as the coating antigen. The samples from each group (6 rabbits) were pooled and then analyzed as one sample in triplicate. Bars represent the mean ± standard deviation (SD) in each group. Different letters (a vs. b, a vs. c, b vs. c) indicate significant differences (*p* < 0.01) at the same time point.

**Figure 6 vaccines-07-00172-f006:**
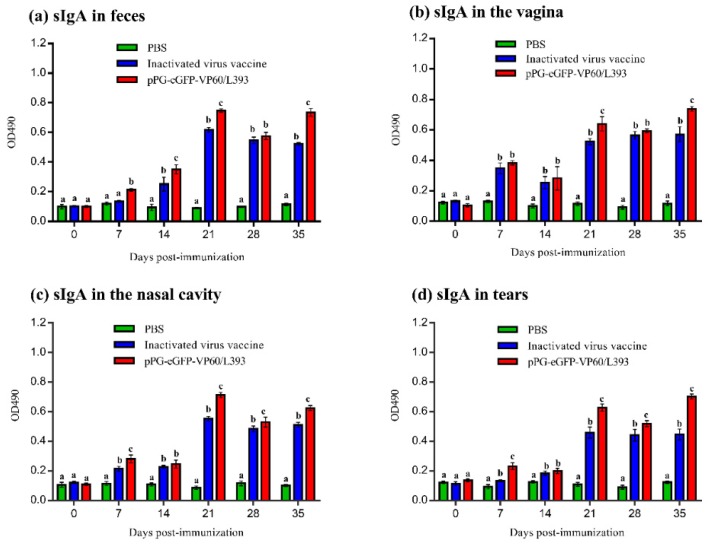
Levels of anti-RHDV-specific IgA in immunized rabbits. Determination of specific anti-RHDV-specific IgA levels in feces (**a**), vagina (**b**), nasal cavity (**c**), and tears (**d**) was performed by ELISA using RHDV as the coating antigen. The samples from each group (6 rabbits) were pooled and then analyzed as one sample in triplicate. Bars represent the mean ± standard deviation (SD) in each group. Different letters (a vs. b, a vs. c, b vs. c) indicate significant differences (*p* < 0.01) at the same time point.

**Figure 7 vaccines-07-00172-f007:**
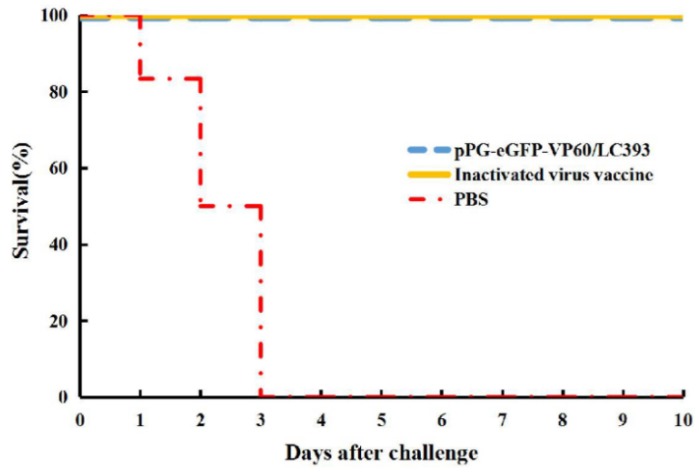
Protective effects in immunized rabbits after challenge with RHDV. The groups administered *L. casei* pPG-eGFP-VP60/LC393, inactivated vaccine, and PBS were infected with 1 mL of RHDV liver crude extract (at RNA copy number of 1.0 × 10^5.97^) by intramuscular injection into the leg at 37 days after the first vaccination. The rabbits were euthanized, and dissection was performed at 10 days post-infection (DPI).

**Figure 8 vaccines-07-00172-f008:**
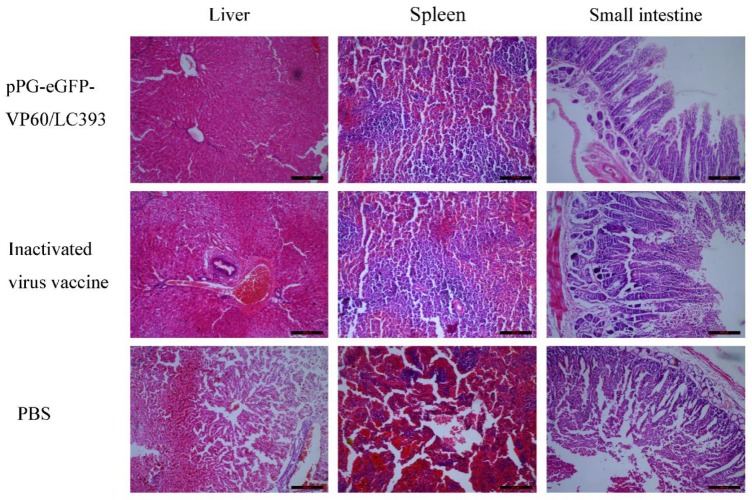
Histopathological evaluation of immunized rabbits. Histopathological examination of H&E-stained liver, spleen, and small intestine. The tissues were obtained from the groups administered *L. casei* pPG-eGFP-VP60/LC393, inactivated vaccine, or PBS at 10 days post-challenge with 1 mL of RHDV liver crude extract (at RNA copy number of 1.0 × 10^5.97^). Scale bar represents 200 µm.

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
