# Peer review of "Recombinant Lactobacillus casei Expressing Capsid Protein VP60 can Serve as Vaccine Against Rabbit Hemorrhagic Disease Virus in Rabbits"

_vaccines, 2019, doi:10.3390/vaccines7040172_

Round 1
Reviewer 1 Report
In the present manuscript, Wang et al. have generated and analyzed a recombinant Lactobacillus casei expressing the major capsid protein of RHDV. They were able to show that administration of this vaccine induced RHDV specific IgG and IgA production and protected rabbits against RHDV infection.
The results are convincing, but the advantage of this vaccine compared to an inactivated vaccine and other vector vaccines delivering the VP60 proteins is not sufficiently described and discussed. There are several mistakes in the text and the information in materials and methods is not sufficient, this is listed in the specific comments.
Specific comments
Title
Lane 3: Text mistake ”.. as vaccine with against RHDV” – what is meant ?
Lanes 26-27: syntax, verb missing “ but present concerns with respect to safety and animal welfare.”
Lanes 27-28: It is not a vaccine against L. casei
Lanes 41, 42 and 55: repetition: “causing considerable economic losses in the rabbit industry/production”
Lanes 52,53: nowadays VP60 is called VP1 and VP10 is called VP2
Lanes 80-82: the meaning of the sentence is not understandable, which results have been confirmed and probably a verb is missing “that the safety and the effectiveness ….”
Lane 95: Is it RHDV-1 or RHDV-2, which RHDV strain was used?
Lanes 96, 97: which study is meant?; a reference is missing
Lane 99: VP60 (VP1) in pMDT18-T, where does the VP60 gene come from? How was the plasmid generated?
Lanes 111, 112: “transformed into LC393 competent cells by electroporation to generate recombinant L. casei”, misleading it is rather “competent LC393 cells”
Lane 129: how was the polyclonal anti VP60 ab prepared?
Lane 130: which gfp antibody from which company was used?
Lane 138: “the expression of recombinant L. casei”, it is rather the expression of VP60 (VP1) and gfp
Lane 140: “rabbit anti-VP60 hyper-139 immune serum“, is this different to the one used for westernblotting (lane 129)
Lanes 150, 153: is 1x 1010 meant or is 1010 correct?
Lanes 152-156: a group with L. casei (without VP1 expression) is missing as a control for the immune stimulating effect of L. casei, the authors should at least discuss the putative antigen independent effect of L. casei in the discussion section
Lane 170: an extra section for the challenge is not helpful but rather confusing as it is the same experiment
Lanes 172-173: how was the viral RNA copy number determined?
Lanes 198- 200: lanes are not understandable, “ass”, which letters indicate what?
Lane 204: competent LC393 cells
Lane 223: PCR specifc for what, primer for which region ?
Lane 228: pPG LC393 was not introduced before
Lane 235: “Confocal microscopy was also used to observe the expression of eGFP in L. casei.” GFP detection via IFA is already included in the sentence before – “expression of the eGFP-VP60 fusion protein” - or was a detection of GFP done without an antibody, at least this was not shown in Figure 4.
Lanes 257-60: it should be named, that there was less IgG present for the L. casei vaccine in comparison to the inactivated vaccine.
Lanes 271, 275: what do the different letters means with regard to significance? Indicate it in the figure legend.
Lanes 291-297: Indicate that or if there is no difference between the inactivated and L. casei vaccine?
Fig 8: mistake in figure heading - spleen
discussion: The authors should discuss the need of several immunization procedures for oral vaccination in comparison to one-time immunization with the inactivated vaccine. Has it been shown that several administrations - on 3 consecutive days, with two boosters – is needed for IgG production and/or protection?
Lanes 394-403 and 399-403 are the same
Reviewer 2 Report
Wang and colleagues report the generation a new orally-administered vaccine for treating RHDV. The manuscript is well-written and the reported findings may provide a cost-effective and safer alternative to immunise farm animals against this pathogen. The methodology seems to be rigorous and I would be happy to accept this manuscript after the authors address a number of subjects.
In my eyes, there is a major limitation of this study that it is not discussed in the manuscript. This limitation is about using animals treated with PBS as positive control instead of animals inoculated with the non-recombinant (or at least the strain transformed with the empty vector) parental L. casei strain. Because of the absence of this control, authors cannot guarantee that the immunisation is not an indirect effect caused by the administration of the probiotic strain rather than the recombinant strain expressing the viral protein. This is not far-fetched at all, given the vast number of publications suggesting that probiotics potentiate mucus layer integrity and immune responses. In figure 3B authors show the expression of the recombinant protein in different generations (up to 40th), and they conclude that there is not major events of recombination, gene deletion or loss of the plasmid. Looking at lines 7 and 8, there is an evident shift in the bands that may suggest an insertion or other genetic event. Did the authors check the recovered plasmid through Sanger sequencing? If not, how the authors can guarantee that the integrity of the cassette is preserved? Is the RHDV epitope expressed on the bacterial surface/membrane? If not, could the authors discuss how the immunisation could be achieved? Authors have to include the scale bar in figure 4b. Also, a merged image showing anti-VP60 and GFP signals would provide better clarity. Figure 8. Caption should provide information about the scale bar. Statistical analysis: Please indicate the post-hoc test used for pairwise comparisons. Typos: Please revise inoculum concentration in lines 150, 154 and 173. The final paragraph is duplicated.
Reviewer 3 Report
Manuscript ID: vaccines-628887
Title: Recombinant Lactobacillus casei expressing capsid protein VP60 can serve as vaccine with against Rabbit Hemorrhagic Disease Virus in rabbits.
The manuscript reports the generation of a genetically engineered recombinant Lactobacillus casei expressing the VP60 capsid protein of RHDV (fused to eGFP), to be used as antigen delivery system by the oral route: an oral vaccine to protect rabbits against RHD. Expression of the recombinant fusion protein eGFP-VP60 was confirmed via Western blotting and immunofluorescence. Subsequently, groups of 6 rabbits were immunized with PBS, conventional inactivated vaccine or the candidate oral vaccine. The immune responses elicited (IgG and IgA) in sera and different mucosal secretions were analysed by ELISA. Finally, the rabbits were challenged with a lethal dose of RHDV. All immunized rabbits (either with conventional or L. casei oral vaccine) were protected against lethal viral challenge.
Main comments
- My main concern regarding the manuscript relates to the originality, novelty and relevance of the results reported. The expression of RHDV VP60 capsid protein in a wide arrange of recombinant expression systems (including: bacteria, yeast, baculovirus, several poxviruses, adenovirus, plants, plant viruses, etc.) has been widely documented in the literature. On the other hand, the use of genetically engineered Lactobacillus casei as a delivery system for oral administration of a wide variety of antigens is also well documented. This is the first time L casei has been used to immunize rabbits against RHD by the oral route.
- A paper based on the development of a vaccine against RHDV should take into account the present epidemiological situation of the disease, since 2010..., when a new RHDV related virus (named RHDV2 or RHDVb, and more recently GI.2), with a distinctive pathogenic profile (affecting young rabbits with less than 2 months or rabbits previously vaccinated with classical RHDV vaccines) emerged in France, rapidly spreading worldwide throughout domestic and wild rabbit populations, apparently displacing classical RHDV strains. The manuscript should provide information about this in the introduction, should mention: Le Pendu et al. (2017) J Gen Virol 98:1658-1666, which describes a proposal for a new unified classification system and nomenclature of Lagoviruses (that has been widely supported and accepted), and importantly, should provide information on the RHDV strain used in this study: whether they used classical RHDV strain or a new variant RHDV2 strain (GI.1 or GI.2 according to the new nomenclature).
- It is a good point that the recombinant L. casei generated was free of antibiotic resistance genes (removal of chloramphenicol resistance gene), in order to avoid biosafety and environmental issues. However, in replacement, the authors used eGFP as reporter gene, thus, the final product expressed was an eGFP-VP60 fusion protein. I think it would have been better if the recombinant L. casei generated only expressed the immunogenic protein (RHDV VP60), or if the expression of eGFP was needed (as selection marker), maybe it would have been advisable to express it independently from VP60 protein. RHDV VP60 can spontaneously assemble into virus-like particles (VLPs) when expressed in different recombinant expression systems, and these VLPs are highly immunogenic (more than soluble VP60 protein). Therefore, expressing a fusion protein eGFP-VP60 (which most likely does not retain the ability to assemble into VLPs) might be detrimental for the efficiency of the candidate vaccine. Although the results obtained with the fusion protein showed complete protection, expression of VP60 alone might be more efficient: i.e. require lower doses, less vaccine administrations (9 vaccine doses were used in this study, including prime and boost). In any case, the fact that the recombinant product expressed is the fusion protein eGFP-VP60 should be made more clear in the manuscript. In the current version this is first indicated in page 7 (line 234), while it should be mentioned in the abstract and the introduction.
- It would have been advisable to include an additional group of rabbits immunized with the L. casei strain used (LC393), not expressing the recombinant protein, as a negative control.
- Regarding the ELISA assays (section 2.7 materials and methods and legends of figures 5 and 6) the authors should explain better the procedure used. Each immunogen tested was administrated to a group of 6 rabbits. It is not clear whether they analysed each sample (1 rabbit) separately (1 triplicate each) and the bars in figures 5 and 6 indicate the mean of the whole group (6 x triplicates), or if the samples from each group (6 rabbits) were pooled and then analyzed as one sample (1 triplicate). Given the small standard deviations (SD) shown, it seems the authors followed the second procedure, but this should be clarified.
Other comments.
- Title: “with” should be deleted.
- Lines 80-83: “The results to date have been confirmed that the safety and the effectiveness of L. casei as the oral vaccine vehicle, which were extensively used in protecting individuals against a variety of pathogens” This sentence is hard to understand and should be modified
- Lines 85-86: “Developing an efficient and safe oral vaccine that can induce strong mucosal and systemic immune responses is critical for effective prevention of RHDV”. This is not true. The authors can say developing an oral vaccine against RHDV is “interesting”, ”desirable”, “advisable”, “convenient”, etc., but not “critical” (RHDV has been fairly good controlled with injectable vaccines in rabitries for almost 30 years).
- Lines 96-97: “in our previous study”. This study is not cited (a reference seems to be missing).
- Line 198: “Ass data” (?) I think it should be “All data”.
- Lines 212-213 and legend Figure 3: Eight samples are mentioned (5th, 10th, 15th, 20th, 25th, 30th, 35th, and 40th generations) but Figure 3 shows nine samples (lanes 1-9). Please correct.
- Lines 399-403: This paragraph is duplicated (lines 394-398). Delete.
- References should be carefully revised, there are some where the journal name is missing (i.e. refs: 24 and 29).
